# Reducing Self-Stigma in People with Severe Mental Illness Participating in a Regular Football League: An Exploratory Study

**DOI:** 10.3390/ijerph16193599

**Published:** 2019-09-26

**Authors:** Álvaro Moraleda, Diego Galán-Casado, Adolfo J. Cangas

**Affiliations:** 1Department of Education, Camilo José Cela University, 28692 Madrid, Spain; amoraleda@ucjc.edu (Á.M.); dagalan@ucjc.edu (D.G.-C.); 2Department of Psychology, Health Research Centre, University of Almería, 04120 Almería, Spain

**Keywords:** severe mental illness, sports, self-stigma, recovery, psychoeducation

## Abstract

For the past 15 years, a regular indoor football competition has been taking place in Madrid (Spain) with 15 teams from different mental health services in the city, in which teams face off weekly as part of a competition lasting nine months of the year. We are not aware of whether a similar competition experience is offered in other cities. The purpose of the present study was to evaluate whether participating in this league, called Ligasame, has an influence on participants’ self-stigma. To do so, the Internalized Stigma of Mental Illness scale (ISMI) was adapted into Spanish and applied to 108 mental health patients, 40% of which participated in Ligasame, and the remainder of which did not. The results obtained reflect significant differences between those participating in Ligasame and those that did not in terms of two specific dimensions related to self-stigma (stereotype endorsement and stigma resistance) and total score. On the other hand, no significant differences were found in terms of other variables, such as patients’ prior diagnosis, age or belonging to different resources/associations. In this article, we discuss the importance of these results in relation to reducing self-stigma through participation in a regular yearly mental health football league.

## 1. Introduction

The fact that people with severe mental illness (hereinafter referred to as ‘SMI’) are one of the most socially stigmatized groups has been widely documented [1,2]. Although there are multiple definitions of SMI, perhaps the best known is that provided by the National Institute of Mental Health [3], where it is indicated that these people have the following characteristics, (i) diagnosis of non-organic psychosis or personality disorder; (ii) duration characterized as involving ’prolonged illness and long-term treatment’ and operationalized as a two-year or longer history of mental illness or treatment; and (iii) disability, which includes dangerous or disturbing social behavior, moderate impairment in work and non-work activities and mild impairment in basic needs [4].

It is a fact that these individuals are usually perceived as aggressive, strange, unpredictable, weak, unproductive, etc. (e.g., [5,6,7]), which particularly hinders their inclusion process, and not only do they have to cope with their primary condition but may also experience the secondary impact of mental health stigma. Consequently, stigma itself has been described as a ‘second illness’ [8,9].

The consequences of this prejudice and the negative effect it has on the individual’s psychological well-being can be long-lasting, even after the remission of psychiatric symptoms [10]. Because of this, people within this group tend to be reluctant to seek out mental health services [11], and generally experience poorer-quality medical care compared to the population that with no such diagnosis [12].

Social stigma also influences the individual’s self-image, or self-stigma, causing them to feel as though they have little chance of recovery [13]. To reduce stigma and self-stigma, different programs exist that focus on promoting interactions with mental health patients, providing information on mental health disorders or encouraging activities created by the patients themselves to raise awareness of mental health [14,15,16].

In this sense, sporting activities in normalized contexts can also be an essential element of social inclusion and the fight against stigma and self-stigma [17]. Until now, studies exploring the effects of sports on mental health have focused on the physical and social benefits of playing sports rather than the reduction of stigma or self-stigma [18,19,20,21].

Moreover, within sports programs, very little research focuses on the specific benefits of football, despite the fact that it is one of the most played sports [22,23,24,25,26,27] and, as stated in Decree 2006/2013 (INI) of the European Parliament [28], is an instrument of social inclusion and multicultural dialogue that plays an important social and educational role. Despite this, we have only found one study focusing on the mental health benefits for patients participating in a regular football league in the North West of England [29].

In our case, *Ligasame*, a regular indoor football league promoting mental health, was created in the Community of Madrid 15 years ago. There are currently 15 teams in the league playing twice a week for the championship. Through active and cooperative methodology, its aim is to empower participants to become active agents in the project. The participants’ concerns and suggestions are consistently taken into and addressed through different collaborative channels, such as inclusion in the disciplinary committee, at official meetings, or among the referees responsible for supervising compliance with the rules of the game [30].

Prior to this project, *Ligasame*’s effect on participants in terms of self-stigma had not been evaluated. Therefore, the central goal of the present study was to analyze whether participation in a regular mental health football league may be connected to lower levels of self-stigma among people with SMI compared to subjects not participating in the league, whether they play sports in their free time or not take part in other physical activity.

## 2. Materials and Methods

### 2.1. Sample

The sample population was composed of 108 people suffering from SMI, who used the rehabilitation resources of the Community of Madrid and family associations. See Table 1 for participants’ diagnosis and the mental health resources that referred them.

An intentional non-probability sampling was used, based on voluntary participation, and was subject to the participation criteria in the aforementioned resources and associations combined with a prior diagnosis (schizophrenia, bipolar disorder, major depressive disorder, personality disorder or obsessive-compulsive disorder), with ages ranging from 20 to 68 years of age ($\bar{x}$ = 41.44; d.t. = 10.05), in which 40% had voluntarily participated in Ligasame for at least one year and 60% had not.

### 2.2. Instruments

Internalized Stigma of Mental Illness scale (ISMI) [31]. The present study uses the version of this scale translated into Spanish by the Andalusian Health Department [32]. The psychometric properties of this Spanish version were validated with good values for the total score of the scale: internal consistency reliability, Cronbach’s α value of 0.91, test-retest, r = 0.95 (*p* < 0.001), and convergent validity confirmed as significant correlations (*p* < 0.001) [33]. This instrument has been translated into more than 40 languages [34]. This self-report is composed of 29 Likert scale items scored on a four-point scale (strongly disagree, disagree, agree, and strongly agree). The instrument measures the subjective experience of stigma through a total score and a structure with five dimensions: alienation, stereotype endorsement, perceived discrimination, social withdrawal, and stigma resistance. The stigma resistance scale scores in reverse; for the calculation of the score of this scale, the score of each item has been subtracted from five.

A registration form designed for this study was used to record and assess data such as a gender, age, diagnosis, participation in *Ligasame*, playing a sport, weekly hours of physical activity, as well as rehabilitation or association resources.

### 2.3. Procedure

An ex post facto research design was used, employing the ISMI questionnaire and the registration form. It was applied by professionals at psychosocial rehabilitation centers in the Community of Madrid and family associations. The participants completed the questionnaire for the first time and signed an informed consent form. Their decision to participate was voluntary and anonymity and confidentiality were guaranteed with regards to data collection and processing. The participants had a personal four-digit password and personal data that could identify them was never requested. The study was carried out in accordance with the Declaration of Helsinki Ethics with the approval of the Research Ethics Committee of the Camilo José Cela University (Spain).

### 2.4. Data Analysis

Data analysis was carried out using the SPSS (version 25.0, IBM Corporation, Armonk, New York, USA) statistical package. We initially performed a descriptive analysis of the sample, and we calculated the mean score and standard deviation of the scales used. Then, we used ANOVA to determine possible differences in the five dimensions and total score of the ISMI questionnaire: by existing diagnosis, prior rehabilitation services accessed, playing sports, and participation in Ligasame. There are no post hoc tests, since in the first three ANOVA there were no significant differences, and in the last one there were only two groups (yes/no participation).

## 3. Results

The first aspect of analysis involved determining whether the participants’ diagnosis implied a difference between means in self-stigma perception subscales, but no statistically significant differences were found (Table 2). In parallel, no correlation was found between the patients’ age and total self-stigma levels (r < 0.200, *p* = 0.150) and each self-stigma subscale (r < 0.200, *p* = 0.235, *p* = 0.097, *p* = 0.136, *p* = 0.361, *p* = 0.878, respectively). Regarding the distribution of users based on their attendance at rehabilitation resources or associations, no evidence was found indicating that a particular resource or association frequented by participants in the study was significantly better than any of the others in terms of reducing self-stigma on any of the subscales (Table 2).

The research variable related to whether or not playing sports and the number of hours dedicated, as identified by the participants themselves, had an impact on a different perception of self-stigma in the total score, as well as on each of the subscales. Following analysis, we could not establish differences with respect to self-stigma between those playing sports (not for Ligasame) and those who did not due to the exclusive effect of sports (Table 2). Similarly, we found no correlation between the number of hours and self-stigma levels (r < 0.200, *p* > 0.100, both with regard to the total score and its dimensions), as well as no differences (*p* > 0.05) in self-stigma based on variables derived from the number of hours using dichotomized grouping (insufficient/sufficient) or grouping by levels (low/medium/high).

The last goal was to determine whether participating in the specific activities of a structured mental health league, namely Ligasame, resulted in different levels of perception of self-stigma (Table 3). Statistically significant differences were found with respect to participation in Ligasame (Table 4), specifically, higher scores in two dimensions, assumption of stereotype or self-stigma (F = 6.588, *p* = 0.012) and stigma resistance (F = 7.031, *p* = 0.009), as well as the total score (F = 5.831, *p* = 0.017).

## 4. Discussion

The main goal of this study was to determine if participating in a regular indoor football league like *Ligasame* could be related to decreased self-stigma among participants. Our result concurs with those of Livingston and Boyd [35] in the lack of statistically significant differences based on the diagnosis type. These authors conducted a meta-analysis (127 articles) of the scientific literature on internalized stigma and found that the diagnosis variable has no significant relationship with internalized stigma.

Moreover, no relationship was found between self-stigma and the resource from which participants had been referred. This is possibly due to the fact that all of these resources and associations carry out local-level actions in their catchment areas and promote broader collaboration with other organizations (municipal and community organizations, social services, mental health services, universities and educational institutions, etc.) directed at personal empowerment and eliminating prejudice among patients and third parties [36].

With regards to the lack of differences between self-stigma and playing a sport in itself, we must not forget that there do exist theoretical trends that present sports as a facilitator of positive values with an important integrative and socializing function [37]. Nevertheless, in our case, sports must be thought of as an instrument to adopt certain orientations and basic conditions for it to be considered from a socio-educational and transformative viewpoint [38]. Playing a sport without forming a part of any specific rehabilitation-focused program is not necessarily associated with the elimination of, or a reduction in, internalized prejudice.

Therefore, the main finding of this study is the differences found with respect to participation in *Ligasame* as they relate to a reduction in self-stigma in two particular dimensions (stereotype endorsement and stigma resistance), as well as the total score. These results reveal that belonging to a specific and structured program with goals closely tied to the inclusive process and the participation of people with SMIs in the community through sports has a positive effect on decreasing internalized prejudice.

Participation in *Ligasame* is not limited to people with SMI, meaning that people without disorders of this kind may also access the program as volunteers or through other collaborative roles (e.g., internships). This case study demonstrates that playing a sport (through *Ligasame*) becomes an effective strategy for offering a positive image of this group of people, while eradicating the ignorance, incorrect beliefs, and ideas that surround them by establishing a normalized relationship in which direct contact is essential [17]. Moreover, *Ligasame* organizes two events a year designed to combat stigma. These inclusive tournaments (Christmas and spring) allow people with and without mental illness to spend the day together, encouraging them to get to know each other, thereby contributing to the promotion of social transformation [30].

Another important aspect to highlight is that the sporting activities are coordinated by mental health professionals, who are also responsible for coaching training and matches. The continued presence of these types of professionals allows *Ligasame* to be directed from a treatment standpoint, incorporating scientific perspectives on recovery geared towards psychiatric rehabilitation [39]. Each team that makes up *Ligasame* is structured based on the individual needs of each of its participants in order to improve different aspects of life (e.g., independence, self-esteem, responsibility, etc.), enabling a more positive and objective assessment of oneself.

With respect to the limitations of the present study, we believe that it is necessary to increase the sample size to be able to compare the variables evaluated in a way that is more reliable. It would also be interesting to evaluate more objective measurements with regards to participation in sports (e.g. intensity, structure of the activity, etc.) to be able to analyze whether participating in a sporting activity outside of Ligasame generates differences in self-stigma. We should consider other aspects as if the time of participation in this league, the pharmacological treatment, the possible side effects of the medication, or the physical condition could influence the results. These are aspects that we want to investigate in more detail in future studies.

Finally, moving towards longitudinal studies by applying a pretest and posttest would be an important step toward the causal inference. This would allow for an evaluation of whether a reduction in self-stigma takes place from the time that participants join Ligasame up until a certain time of participation in the league or until they decide to leave it.

## 5. Conclusions

The context in which the activity takes place is also beneficial for participants. By hosting these activities within a community environment, using facilities shared with a wider population, outside of the strictly clinical context or a context exclusively focused on their illness, the individual can develop a more normalized perception of the activity itself and their own involvement in it [40].

Moreover, it is important to note that *Ligasame* participants are in constant contact with other people with similar disorders. This contact expands to the national and international level when Ligasame participants attend events organized by other Spanish autonomous communities or different countries. These experiences allow participants to build social relationships and discover new bonds of friendship and mutual support, thus creating a space to exchange experiences [40]. Through this contact, personal strategies are directed at positive reinforcement and empowerment to help cope better with the consequences of mental illness, trying to eliminate stereotypes, as well as inaccurate and harmful self-perception. Thus, the result of the current research reflects the importance of this regular football league to reduce internalization prejudices, which is an important rehabilitative element.

## Figures and Tables

**Table 1 ijerph-16-03599-t001:** Distribution of participants by existing diagnosis and prior rehabilitation services accessed.

Factor	Type	n	%
Existing diagnosis	Major depression	4	3.7
	Schizophrenia	84	77.8
	Bipolar disorder	10	9.2
	Personality disorder	7	6.5
	Obsessive compulsive disorder	3	2.8
Rehabilitation resources	Associations	8	7.4
	Day Centres	5	4.6
	Vocational Rehabilitation Centre	55	50.9
	Psychosocial Rehabilitation Centre	27	25.0
	Mini-residence	11	10.2
	Supported Accommodation	2	1.9

**Table 2 ijerph-16-03599-t002:** Effects of the analysis of variance by existing diagnosis, by use of rehabilitation resources and by playing sports (yes/no) (SS: sum of squares; MS: mean of squares; df: degrees of freedom).

Dependent Variable: Diagnosis	SS	df	MS	F	*p*-value	ηp2
Alienation	139.671	4	34.918	1.981	0.103	0.071
Assumption of stereotype or self-stigma	107.297	4	26.824	1.232	0.302	0.046
Perceived discrimination or experienced discrimination	99.689	4	24.922	1.831	0.129	0.066
Social isolation	101.465	4	25.366	1.390	0.243	0.051
Resistance to stigma	53.795	4	13.449	1.227	0.304	0.045
Total	1255.121	4	313.780	1.380	0.246	0.051
Alienation	87.526	5	17.505	0956	0.448	0.045
Assumption of stereotype or self-stigma	205.000	5	41.000	1.950	0.093	0.087
Perceived discrimination or experienced discrimination	86.929	5	17.386	1.253	0.290	0.058
Social isolation	109.438	5	21.888	1.193	0.318	0.055
Resistance to stigma	94.210	5	18.842	1.765	0.127	0.080
Total	2219.689	5	443.938	2.017	0.082	0.090
Alienation	0.982	1	0.982	0.053	0.818	0.001
Assumption of stereotype or self-stigma	3.479	1	3.479	0.157	0.693	0.001
Perceived discrimination or experienced discrimination	25.234	1	25.234	1.812	0.181	0.017
Social isolation	2.577	1	2.577	0.138	0.711	0.001
Resistance to stigma	34.810	1	34.810	3.214	0.076	0.029
Total	11.267	1	11.267	0.048	0.826	0.000

Significant differences: * *p* < 0.05, ** *p* < 0.01.

**Table 3 ijerph-16-03599-t003:** Descriptive statistics by participation in Ligasame (M: Mean; SD: Standard deviation).

Factor	Type	N	M	SD
Alienation	No	65	2.99	0.73
	Yes	43	2.79	0.67
	Total	108	2.91	0.71
Assumption of stereotype or self-stigma	No	65	3.23	0.62
	Yes	43	2.90	0.69
	Total	108	3.10	0.67
Perceived discrimination or experienced discrimination	No	65	2.90	0.75
	Yes	43	2.78	0.74
	Total	108	2.85	0.75
Social isolation	No	65	3.00	0.71
	Yes	43	2.80	0.72
	Total	108	2.92	0.72
Resistance to stigma	No	65	2.70	0.67
	Yes	43	2.36	0.60
	Total	108	2.56	0.67
Total	No	65	2.98	0.51
	Yes	43	2.74	0.52
	Total	108	2.89	0.52

**Table 4 ijerph-16-03599-t004:** Effects of the analysis of variance by participation in Ligasame (yes/no). (SS: sum of squares; MS: mean of squares; df: degrees of freedom).

Dependent Variable: Participation in Ligasame (y/n)	SS	df	MS	F	*p*-value	ηp2
Alienation	38.364	1	38.364	2.122	0.148	0.020
Assumption of stereotype or self-stigma	137.522	1	137.522	6.588	0.012 *	0.059
Perceived discrimination or experienced discrimination	9.107	1	9.107	0.647	0.423	0.006
Social isolation	40.304	1	40.304	2.201	0.141	0.020
Resistance to stigma	73.585	1	73.585	7.031	0.009 **	0.062
Total	1286.315	1	1286.315	5.831	0.017 *	0.052

Significant differences: * *p* < 0.05, ** *p* < 0.01.

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
