# Peer review of "Reducing Self-Stigma in People with Severe Mental Illness Participating in a Regular Football League: An Exploratory Study"

_ijerph, 2019, doi:10.3390/ijerph16193599_

Round 1

Reviewer 1 Report

Interesting article regarding the quality of life of patients with a mental disorder is concerned. The article has a good approach and uses a validated tool that reinforces the results found and the conclusions they confirm. It has a correct statistical treatment of the data and these are reliable.

Some minor clarifications:

-The procedure for the validation of the survey in Spanish (Cronbach's α), should be placed in the statistics section of materials and methods that are purely statistical procedures performed by the authors to validate the survey and its results are relevant.

- It is not clear to me if in the 15 years of study, the participants have been surveyed once after at least one year of sport and if there are participants with 15 years of activity in the LIGASANE. It should be clear in the methodology the average of years of sports activity and if they did the questionnaire once or every year of sports practice.

Author Response

1) The procedure for the validation of the survey in Spanish (Cronbach's α), should be placed in the statistics section of materials and methods that are purely statistical procedures performed by the authors to validate the survey and its results are relevant. The procedure for the validation of the survey in Spanish (Cronbach's α), should be placed in the statistics section of materials and methods that are purely statistical procedures performed by the authors to validate the survey and its results are relevant.

It has been clarified in the article that we have used in the study the validated version of the scale in Spanish by Bengochea-Seco et al. 2016 (Line 88). Since it is an instrument already validated in Spanish, we have not analyzed the psychometric properties of the test. In addition, the sample used in the study is not very large, so it is not suitable for studies on reliability or validation.

2) It is not clear to me if in the 15 years of study, the participants have been surveyed once after at least one year of sport and if there are participants with 15 years of activity in the LIGASANE. It should be clear in the methodology the average of years of sports activity and if they did the questionnaire once or every year of sports practice.

It is specified in the text, which should have participated in Ligasame at least one year (Line 81).

This was the requirement to participate in the study, but the average time taken was not measured, which has been introduced as a limitation of the study (all limitations are moved to the point of discussion to clarify the information) and the interest of investigating this variable in future works (Line 191). For this reason, it has also been added in the entitle that it is an exploratory study (Line 3).

Reviewer 2 Report

This is a potentially interesting study about the comparison of self-rated self stigma levels of  people with severe mental illness participating to a football league vs people with severe mental illness who do not 

The following issues need to be addressed:

1) English needs revision; some sentences and expressions are odd (e.g. line 200 "executed" ; line 202 "de manuscript";  "supervised apartment" do you mean supported accommodation, residential care or both?

2) It is unclear if the validated Spanish version of the ISMI has been used:  10.1016/j.rpsmen.2016.01.009 , if yes it should be cited, if not one would want to know why and this should be acknowledged in the limitations;

3) The ISMI should be much better explained in the method section, for example it should be specified that possible answers are on a four point scale where strongly disagree correspond to 1 and strongly agree correspond to 4, whereby higher scores correspond to higher self stigma. As far as I know the resistance to stigma subscale is reverse scored, therefore higher scores also correspond to higher self stigma;

4) Therefore, looking at your Table 3 the group which participated to Ligasame (Yes) has significantly HIGHER global self stigma, and assumption of stereotypes and resistance to stigma subscales scores, just the opposite of what you say in the Discussion. You need to check this properly. This Table is very unclear, figures are too close, and mean scores should be reported in a scale from 1 to 4;

5) Even if the Ligasame group had lower scores than the no football one the methodology of the study does not allow to say that "this article has evidenced a reduction of self-stigma" in the participants (line 196), this could only be achieved in a longitudinal study with pre- and post- intervention measures. What you could say, if it were true (that is, if self stigma scores in the ligasame group were lower and not higher as in your Table 3 reports now) is that the ligasame group shows lower self stigma than the other group but no causal inference at all can be drawn by your design. This should be acknowledged as a limitation

Author Response

1) English needs revision; some sentences and expressions are odd (e.g. Line 200 "executed"; Line 202 "de manuscript"; "supervised apartment" do you mean supported accommodation, residential care or both?

The text has been revised by a new native English person to correct possible odd expressions. The specific expressions indicated by the reviewer have also been changed. (table 1).

2) It is unclear if the validated Spanish version of the ISMI has been used:  10.1016/j.rpsmen.2016.01.009 , if yes it should be cited, if not one would want to know why and this should be acknowledged in the limitations

This matter has been clarified in the text (Line 88), indicating that a version adapted and validated to Spanish has been used, by Bengochea-Seco el al, 2016 DOI:10.1016/j.rpsmen.2016.01.009. 

3) The ISMI should be much better explained in the method section, for example it should be specified that possible answers are on a four point scale where strongly disagree correspond to 1 and strongly agree correspond to 4, whereby higher scores correspond to higher self stigma. As far as I know the resistance to stigma subscale is reverse scored, therefore higher scores also correspond to higher self stigma,

The characteristics of the scale have been better explained in the method section.

It has also been clarified all the items are corrected in ascending order, especially in the "stigma resistance" dimension that scores in reverse. To calculate the score of this dimension, the score of each item has been subtracted from five (Line 92-94).

4) Therefore, looking at your Table 3 the group which participated to Ligasame (Yes) has significantly HIGHER global self stigma, and assumption of stereotypes and resistance to stigma subscales scores, just the opposite of what you say in the Discussion. You need to check this properly. This Table is very unclear, figures are too close, and mean scores should be reported in a scale from 1 to 4.

In table 3 we have corrected the scores of the dimensions and total score, so that they all score increasingly and are scaled between 1-4. In this way, it is verified that the results are in line with the discussion.

5) Even if the Ligasame group had lower scores than the no football one the methodology of the study does not allow to say that "this article has evidenced a reduction of self-stigma" in the participants (Line 196), this could only be achieved in a longitudinal study with pre- and post- intervention measures. What you could say, if it were true (that is, if self stigma scores in the ligasame group were lower and not higher as in your Table 3 reports now) is that the ligasame group shows lower self stigma than the other group but no causal inference at all can be drawn by your design. This should be acknowledged as a limitation.

This sentence has been deleted in the text. Also, the lack of a longitudinal design that allows the causal inference is included in the limitations (Line 194) (all limitations are moved to the point of discussion to clarify the information).

Reviewer 3 Report

The study examines whether patients’ with severe mental illness participation in football league competitions affect participants' self-image. Results are promising. The paper is relevant as it reinforces the importance of stigma-reducing interventions/programs like participation in sporting activities in patients with severe mental illness. The paper is well written. Authors have listed the limitations of the study which does limit the generalizability of the study findings.

To improve the quality (to reduce the potential biases) of the study, it would be helpful to know:

If subjects with SMI were receiving psychopharmacology treatment (could be source of poor self-image) The reasons for non-participation in the study, which could be due to poor self-image If poor physical health status and potential side effects of medications were affecting non-participation in Ligasame (potential baseline poor self-image).

Author Response

If subjects with SMI were receiving psychopharmacology treatment (could be source of poor self-image) The reasons for non-participation in the study, which could be due to poor self-image If poor physical health status and potential side effects of medications were affecting non-participation in Ligasame (potential baseLine poor self-image).

This idea has been incorporated as limitations of the study (all limitations are moved to the point of discussion to clarify the information) and it will be important to investigate its effect in future research (Line 191).

Reviewer 4 Report

Overall, I found this study to be interesting. I recommend that editing be done to strengthen the paper. For example, some sentences are rather long-winded and can be tightened for better coherence. 

There must be a definition of severe mental illness and the cluster or spectrum of mental disorders that fall within this definition. At the moment, the paper assumes that the reader has prior knowledge and this may not be the case for everyone who reads it. 

In Line 29, the authors write about people living with SMIs being stigmatised as "widely documented", but there are no references that substantiate this claim. In other words, this sentence must be adequately referenced. 

In line 39, the authors write about a "second illness". If the consequences are equivalent to second illness, then this should be better integrated. 

Line 35: "...compared to the general population" - when they do utilise mental health services?

Line 42: including self-stigma?

Line 43: "...playing sports rather than [reducing/eradicating] stigma...?

Line 46: Should the argument here not be linked to football and the benefits to good mental health? There should be a stronger argument here. 

Line 63-70: restructure this paragraph for better coherence. 

The tables in the document should be displayed as per IJERPH standard format. Please avoid centering text in tables. 

Line 90: "Anonymity and confidentiality...were guaranteed..." How was this achieved in the study?

Line 116: Omitting analyses because no correlation, but then there are analyses described within that sentence. It is a bit confusing - if it answers the objectives it should be shown in the paper. 

In the discussion, how does self stigma and internalised stigma differ or are the similar concepts?

Author Response

1) I recommend that editing be done to strengthen the paper. For example, some sentences are rather long-winded and can be tightened for better coherence. 

The text has been edited to confirm a better coherence.

2) There must be a definition of severe mental illness and the cluster or spectrum of mental disorders that fall within this definition. At the moment, the paper assumes that the reader has prior knowledge and this may not be the case for everyone who reads it. 

A definition of the term severe mental disorder and its characteristics has been included (lines 31-37).

3) In Line 29, the authors write about people living with SMIs being stigmatised as "widely documented", but there are no references that substantiate this claim. In other words, this sentence must be adequately referenced. 

We have added some references to corroborate this opinion

4) In Line 39, the authors write about a "second illness". If the consequences are equivalent to second illness, then this should be better integrated. 

The meaning of this concept has been better explained and the reference of the authors who proposed it has been added (Line 38-40).

5) Line 35: "...compared to the general population" - when they do utilise mental health services?

We have eliminated the expression “general population” and clarified that it is the population that has not received this diagnosis (Line 44)

6) Line 42: including self-stigma?

We have incorporated in the phrase: stigma and self-stigma

7) Line 43: "...playing sports rather than [reducing/eradicating] stigma...?

This error has been corrected 

8) Line 46: Should the argument here not be linked to football and the benefits to good mental health? There should be a stronger argument here.

We have added a reference from the European Parliament to strengthen the argument about positive effects of football (Line 55-57)

9) Line 63-70: restructure this paragraph for better coherence

This paragraph has been restructured (Line 74-81)

10) The tables in the document should be displayed as per IJERPH standard format. Please avoid centering text in tables

The format of all tables has been reviewed and modified

11) Line 90: "Anonymity and confidentiality...were guaranteed..." How was this achieved in the study?

We have clarified how we guarantee the anonymity and confidentiality of the participants in the study (Line 104).

12) Line 116: Omitting analyses because no correlation, but then there are analyses described within that sentence. It is a bit confusing - if it answers the objectives it should be shown in the paper.

This sentence has been changed to avoid confusion

13) In the discussion, how does self stigma and internalised stigma differ or are the similar concepts?

Both terms are synonyms and are used to prevent word repetition

Round 2

Reviewer 2 Report

I am satisfied that Authors have addressed all the issues highlighted in my previous review